# Transforming Circe: Latin Influences on the Depiction of a Sorceress in Renaissance *Cassone* Narratives

Margaret Franklin

James Pearson Duffy Department of Art and Art History, Wayne State University, Detroit, MI 48202, USA; ai4589@wayne.edu

**Abstract:** This article addresses the use of Latin accounts of Homer's archetypal sorceress, Circe, in visual narratives constructed to embellish quattrocento marriage chests (cassoni). I argue that Apollonio di Giovanni employed the writings of both ancient (Virgil) and late medieval (Boccaccio) Latin authors to construct a characterization of Circe that rendered her power to transform men into beasts relevant to the functioning of Early Renaissance homes and societies.

**Keywords:** Circe; Odysseus; cassoni; Homer; Virgil; Boccaccio; Renaissance; marriage; gender roles; women

The obstacles that Homer's Odysseus (Latin 'Ulixes,' English 'Ulysses') confronts on his journey home from the Trojan War include two fantastical beings determined to rob him and his companions of their lives: a cyclops who wants to eat them and a sorceress who wants to turn them into swine. Polyphemus's anthropophagy is in response to a perceived wrong; the men had, uninvited, settled down in his cave and eaten his cheese. His retribution for this incursion, while plainly a greater violation of hospitality than that committed by the Greeks, can nonetheless be understood as a barbaric form of eye-for-an-eye justice. Circe's objective in changing the forms (but not the minds) of guileless men into that of beasts is, however, obscure; she welcomes the Greeks into her palace only to dispassionately deprive them of their discernible humanity. Her response to Odysseus' ultimate thwarting of her intentions is also puzzling—she restores his men and becomes his generous and knowledgeable benefactor. Circe's apparently capricious inclinations and largely unassailable power to destroy men inspired later writers to explicate the motivations of this enigmatic being. Latin writers, most notably Virgil, Ovid, and their medieval commentators, fashioned an enchantress who used her powers towards altogether malevolent ends. The following study argues that Apollonio di Giovanni, master of a thriving 15th-century Florentine *cassone* workshop, selectively incorporated the work of influential Latin authors into a characterization of the Odysseus/Circe story that spoke to Renaissance anxieties regarding the roles of the sexes both as marital partners and as contributors to well-functioning societies.

Early Renaissance enthusiasm for Homer's epics was fueled largely by a veneration of Virgil, who wielded unrivalled *auctoritas* over early humanists. Although the *Aeneid*, known to be indebted to the *Iliad* and *Odyssey*, was believed to be superior in every way to its antecedents, scholars remained desirous of reading Homer's work.[1] Unfortunately, a language barrier frustrated early humanists in their attempts to access the text; in 1440 Florentine Chancellor Leonardo Bruni lamented "for 700 years now, no one in Italy has been able to read Greek." (Bruni 2010, p. 812). The best Western European access to Homer was found via Leontius Pilato's widely denounced *ad verbum* Latin translation of the epics (c.1362), which Coluccio Salutati (1392) deemed "barbarous and unsophisticated" (Sowerby 1997, p. 57).[2] Circe would therefore have been best known to Renaissance readers through two ancient Latin literary works, Virgil's *Aeneid* and Ovid's *Metamorphoses*, and the late medieval biographies of Giovanni Boccaccio (*De mulieribus claris* and *Genealogia deorum*

*gentilium*) that aimed to synthesize ancient accounts of legendary and historical figures into morally meaningful *vite*.

Virgil's treatment of the Circe legend is brief; Aeneas and his Trojans do not actually meet the sorceress but sail past her "cruel coasts" (*Aen.* VII, 28), where they hear the growls, howls, and roars of animals "whom the savage goddess Circe changed, by overwhelming herbs, out of the likeness of men into the face and form of beasts" (*Aen.* VII, 23–25).[3] Ovid renders a fuller reimagining of Homer's tale which, although told from the perspective of one of Odysseus' companions rather than Odysseus himself, incorporates the narrative essentials of its source material.[4] In both Homer and Ovid the Greeks land on Circe's island, where a reconnaissance team is welcomed into her palace. After the men eagerly guzzle her proffered drug-laced concoction, she uses her wand to transform them into despairing pigs and confines them in pens. A sole unbewitched scout escapes to inform Odysseus of their fate and, striking out for the palace, the hero encounters Hermes (Latin 'Mercurius,' English 'Mercury'), who not only gives him a magical plant called 'moly' that will enable him to resist Circe's blandishments, but detailed instructions on how to frighten and subdue her.

Ovid establishes the enchantress's motives for the unprovoked assault on Odysseus' men by recounting the Roman tale of her metamorphosis of King Picus of Latium into a woodpecker. In this non-Homeric myth, the sorceress is driven by lust, jealousy, and vengeance. Her frantic pursuit of the faithful husband involves the conjuring of hallucinogenic deceptions—a phantom boar, a blinding fog. When her dark arts prove ineffectual, she not only transforms the king into a bird (and his men into wild beasts) but commemorates her 'victory' by erecting a hybrid man-bird statue in her garden. Virgil also alludes to Picus's fate and likewise attributes it to Circe's lust (*Aen.* 247–52). Yet another example of the ruinous consequences of Circe's intemperate ardor is integrated into the *Metamorphoses*: upon being rejected by the sea god Glaucus, the merciless goddess uses potions and spells to transform the legs of his beloved Scylla into a pack of vicious dogs. Ovid's Scylla is not the six-headed beast who consumes six of Odysseus' men in the *Odyssey*, but a post-Homeric version. Nonetheless, these additional stories of transformation helped to seal, for Latin readers, Circe's essential nature as dangerously lubricious.

Although Virgil and Ovid, like Homer embedded Circe in a mythological world where her powers are supernatural (she is the daughter of the sun god Helios), other ancient writers familiar to early humanists historicized her in the form of a mortal seductress. Diodorus Siculus, a first-century Greek historian known in the quattrocento through translation, identified her as the cruel queen of the Scythians who poisoned her husband, usurped his throne, and committed "many cruel and violent acts against her subjects" (Oldfather 1933) before fleeing to a desert island.[5] Servius, whose fifth-century Latin commentary on the *Aeneid* was admired by Renaissance scholars, is directly quoted by Boccaccio in *De mulieribus claris* to support the claim that Circe was a "famous courtesan" (*Commentarius in Vergilii Aeneidos libros* 7.19.2; *Dmc* 38.6). Whether or not they argued for her historicity, medieval Latin commentators viewed Circe's power over men as allegorical of the wages of lust. The anonymous Third Vatican Mythographer, for example, observes that "by her lust and blandishments [she] led men from a human way of life to that of wild animals so that they might give their attention to lust and pleasures." (Pepin 2008, p. 305). Others, like the Second Vatican Mythographer, extend the allegory to read Odysseus' immunity as representative of the wisdom necessary to combat female temptresses: "blameless Ulysses ignored her because wisdom despises lust." (Pepin 2008, p. 300).

Boccaccio assumed a euhemeristic stance when he wrote his widely influential compendium on the lives of 106 exceptional females that were mostly drawn from the myths, history, or legends of Greek or Roman antiquity. *De mulieribus claris* (1361–1374) not only became an authoritative source for Renaissance scholars and literary and visual artists, but was translated into European vernacular languages and heavily informed Chaucer's *The Legend of Good* Women (1386–1394) and Christine de Pizan's *Le Livre de la cité des dames* (1405).[6] Boccaccio's stated goal in writing *De mulieribus claris* was a desire to provide

models—both exemplary and cautionary—of women who might "drive the noble towards glory and to some degree restrain villains from their wicked acts" (*Dmc* Pref.7).[7] His biographies were largely shaped as object lessons for the benefit of his contemporaries; like other euhemerists, he fashioned the gods and demi-gods of antiquity as eligible for emulation or repudiation by converting their extraordinary actions into recognizable behavior attributable to human strengths and failings. He devotes many pages to Circe, a woman who was "forceful and eloquent but not overly concerned with keeping her chastity untarnished so long as she got something she wanted" (*Dmc* 38.5). Under the influence of her "wiles and charming words," men lost their reason; some she "pushed into robbery and piracy; others she induced with her tricks to cast all honor aside and take up commerce and trading" (*Dmc* 38.5). Boccaccio soberly observes that "If we consider human behavior, we see plainly enough from this instance that there are many Circes everywhere, and many more men whose lust and vice change them into beasts" (*Dmc* 38.6).

The only known late trecento/early quattrocento visual depictions of Circe are manuscript miniatures painted to accompany portions of text in *De mulieribus claris* and its translations. These paintings emphasize the sexualized characterization of Circe and her victims to the complete neglect not only of Boccaccio's expanded critique of the fates of men who cannot resist her allure, but the *Odyssey* narrative itself. A miniature from a French translation of *De mulieribus claris* (*Le livre de femmes nobles et renomées*, Figure 1) depicts Circe as welcoming the embraces of at least one, and possibly several, admirers. She gestures towards three large, circling birds that suggest the sub-human transformation that awaits those who succumb to her charms. This image encapsulates a general sense of degradation associated with the indulgence of lecherous impulses; unlike the hapless Greeks, the intent of these anonymous men is salacious, and they may therefore be viewed as deserving of what is about to happen to them.

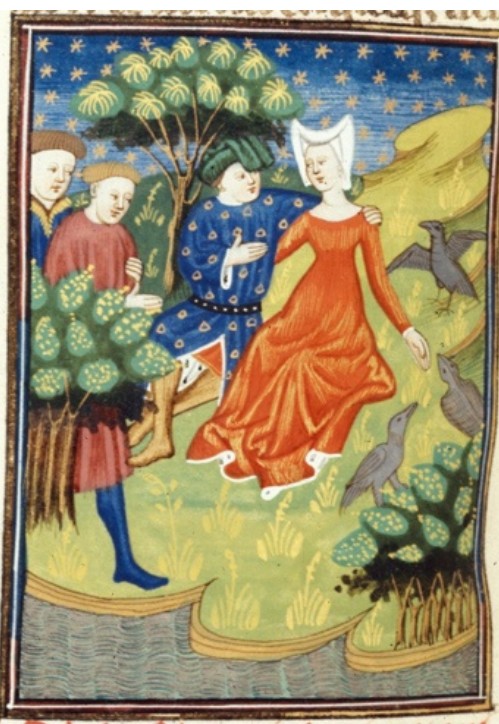

**Figure 1.** Talbot Master, *Circe*, *Le livre de femmes nobles et renomées*, Royal MS 16 G V, f. 42v, c.1440, British Library (public domain photograph).

Sexual desire is even more strongly accentuated in a miniature from yet another anonymous French translation of *De mulieribus claris* (Figure 2). Here, two of the four men who press yearningly towards Circe are represented as hybrid man-beasts. The source of the miniaturist's inspiration of revealing the dual nature of these men is unclear; ancient

texts describe fully transformed beasts who are visually indiscernible from their natural counterparts. Odysseus/Circe narratives that feature hybrids with nude human bodies and animal heads do appear, however, on many excavated Greek pottery vases, suggesting that late medieval illustrators were familiar with unidentified remnants of this ancient method of conveying the lingering humanity of Circe's victims.[8]

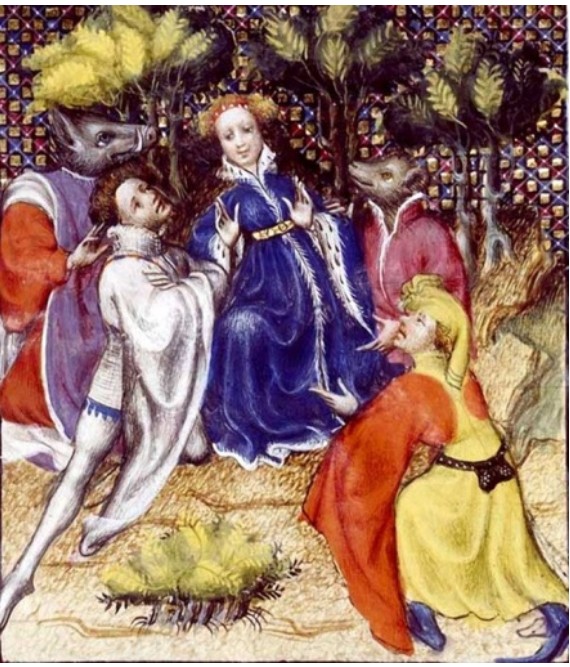

**Figure 2.** Anonymous, *Circe*, *Le livre de femmes nobles et renomées*, Bibliothèque nationale de France, fr. 598, fol. 54v., c.1361–1362 (public domain photograph).

There is no known evidence to suggest that Apollonio di Giovanni had access to visualizations of Circe through manuscripts or other media.[9] Whether or not such imagery was available to the painter, he clearly does not perpetuate Circe's reputation as a lascivious mesmerizer; in *cassone* panels she is modestly dressed from head to toe and is not shown interacting with hybrids or lustful men. She does not even acknowledge the beasts that prowl on her island. What role, then, is she and her story meant to play in narratives that predate by at least half a century the reimaginings of pagan enchantresses that followed the late fifteenth-century shift away from equivocal attitudes towards occult magic and its practitioners?[10] Apollonio's Circes differ distinctly from sixteenth-century Christian European images that, reflecting a zeitgeist that linked witchcraft with demonic sexual practices, accentuate both her seductiveness and the trappings of her black magic.[11] In a 1563 edition of Ovid's *Metamorphosis* (Figure 3), for example, she is both a provocatively clad seductress and a wand- and potion-wielding sorceress.

*Cassoni*, large wooden chests commissioned in pairs for betrothed children, were staple furnishings in Tuscan patrician homes.[12] Made to first carry precious dowry goods to the bride's new home and then to subsequently store linens, clothing, etc., they were typically decorated with painted front and end panels. Consensus holds that the subjects of narrative panels painted to decorate *cassoni* typically derived from Greek or Roman legend or history and were meant to either problematize or reinforce contemporary social norms and assumptions.[13] To this end, *cassone* painters selectively condensed, adapted, and/or amalgamated textual sources with the objective of edifying newlyweds and their households (see, for example, Baskins et al. 2008; Franklin 2006; Baskins 1998; Miziolek 1996; Morrison 1992). The panels that survive, usually having been removed from their chests, represent a minute fraction of those constructed, and similarities between extant *cassone* paintings indicate that

workshops made a practice of reusing popular templates. The complexity of Apollonio's few surviving *Odyssey* narratives, as well as a later panel that imitates Apollonio's composition, bespeaks a strong likelihood that they are all representative of a large, lost, body of comparable work.[14]

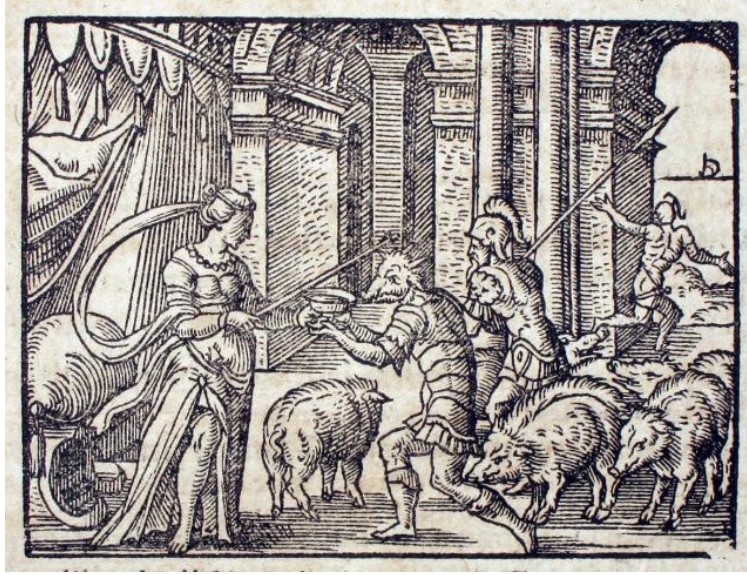

**Figure 3.** Virgil Solis, Circe, Ovid, *Metamorphoses* (trans. Johann Posthius), woodcut, 1563 (public domain photograph).

The first of a well-preserved pair of *Odyssey* pendant panels now in Kraków (Figure 4) prominently features two highlights of the epic's *apologoi*, the hero's adventures with Polyphemus and with Circe.[15] The latter section begins with a prominently foregrounded image of Odysseus' meeting with the messenger god Hermes, who intercepts him before he can reach Circe's palace. In his right hand Hermes carries moly, the plant which will act as a prophylactic against Circe's magical herbs, and in his left the caduceus, variously interpreted as a signifier of wisdom, eloquence, and cunning—all of which will be necessary to neutralize Circe's powers.[16] His concern is not that Odysseus will fall into depravity if he succumbs to his desire for Circe but that, if he does so, she will enfeeble and thereby control him. The god directs him not to "refuse the bed of the goddess" (*Od*. X.297) but cautions him to first "bid her swear the great oath of the blessed gods that . . . she will not make you weak and unmanned, once you are naked" (*Od*. X.299–301).[17] The fear of emasculation is later expressed by Odysseus himself when he makes Circe swear that she won't make him "a weakling, unmanned" (*Od*. X.341). Apollonio's emphasis on Hermes' warning reflects Boccaccio's admonition that the threat presented by Circe-like women extends beyond the temptation to over-indulge in sensual pleasures. The angle at which Hermes holds the caduceus leads the viewer's eye from the foreground back to Circe, creating a visual connection between the two immortals that augurs Odysseus' triumph; by acting on the god's advice, Odysseus will leave the goddess, in every sense, empty-handed.

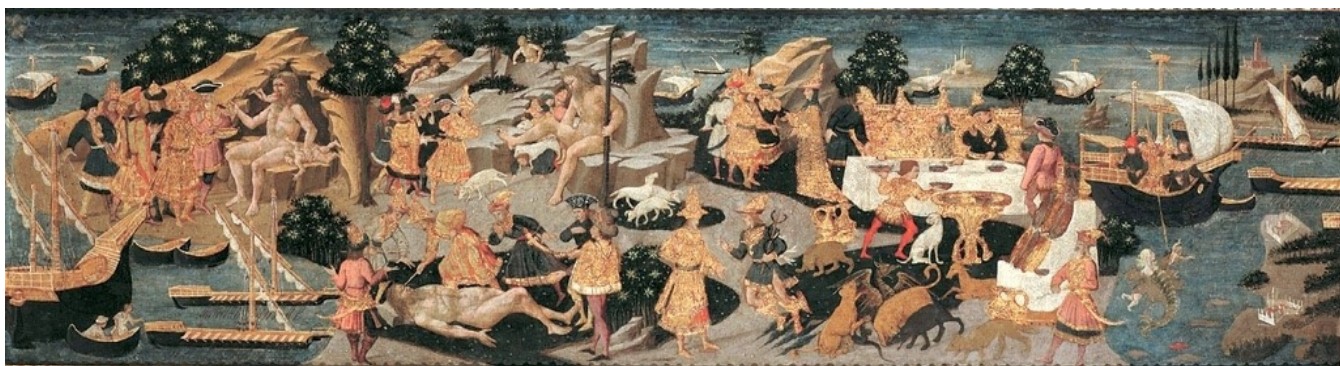

**Figure 4.** Apollonio di Giovanni, *The Adventures of Odysseus*, ca. 1430–1440, tempera on panel *cassone* front, Kraków, Royal Wawel Castle (public domain photograph).

Although the omission of Circe's wand and herbs visually diminishes the power she holds relative to Hermes, recognition of the peril she represents to men who are not aware of her stratagems is, nonetheless, strongly in evidence. In the right center foreground Apollonio depicts a cluster of beasts that Miziolek, in his excellent seminal work on the Kraków panels, identifies broadly as "Ulysses's companions being turned into beasts." (Miziolek 2006, p. 61). However, none of the creatures depicted actually resemble the caged, bristled boars into which, as described by Homer and Ovid, the hero's men were transformed. It is implausible that the only grouping of tame animals on the panel, three unpenned white sheep located in the center of the panel, are associated with Circe—they belong to the same species as those emerging from Polyphemus' cave to graze, and their proximity to his staff connects them to him and his story. It appears, instead, that Apollonio excluded the metamorphosed comrades that trigger Odysseus' entry into the Circe narrative in favor roaring, biting beasts that include a bear, two lions, two canids, and a dragon (Figure 5).[18] Not only are these animals not Odysseus' comrades, but they diverge markedly, both in variety and level of aggression, from the harmless creatures that greeted the Greeks upon their arrival at the sorceress's palace: "lions, and wolves of the mountains, whom the goddess had given evil drugs and enchanted, and these made no attack on the men, but came up thronging about them, waving their long tails and fawning, in the way that dogs go fawning about their master" (*Od*. X.212–216). Ovid's account is comparable; although the men see "lions, bears and wolves, hundreds of them together . . . (they) meant no single scratch of harm. No, they were gentle, and they wagged their tails and fawned on us."[19] (Ovid 1955, p. 318). Odysseus himself never mentions seeing wild beasts, who play no role in the story once he has entered it; as C. M. Bowra observes, their function is to "reveal something sinister in Circe's dwelling, and when they have done that, they are forgotten." (Bowra 1988, p. 56). Apollonio's compositionally prominent beasts plainly serve a different, unforgettable function, as they eternally prowl between the viewer's gaze and Circe's domain.

Of the factors that may account both for the painter's introduction of menacing erstwhile men and his omission of domesticated pigs, Virgil's decision to do the same may have been paramount. There is no mention of tame animals in Virgil's description of Circe's island—only angry, snarling brutes. An illustrious early humanist precedent for adopting Virgil's account of Circe's beasts is found in Boccaccio's sweepingly influential mythography *Genealogia deorum gentilium* (*Genealogy of the Pagan Gods*, begun c.1350). In a chapter devoted to Circe, he perpetuates Virgil's vision of the sorceress's domain by directly quoting the *Aeneid*:

> The raging groans of lions fill her palace –
>
> they roar at midnight, restless in their chains –
>
> and growls of bristling boars and pent-up bears,
>
> and howling from the shapes of giant wolves:

all whom the savage goddess Circe changed,

by overwhelming herbs, out of the likeness

of men into the face and form of beasts.

(*Aen*. VII.19–22; *Gdg* 4.14.2)[20]

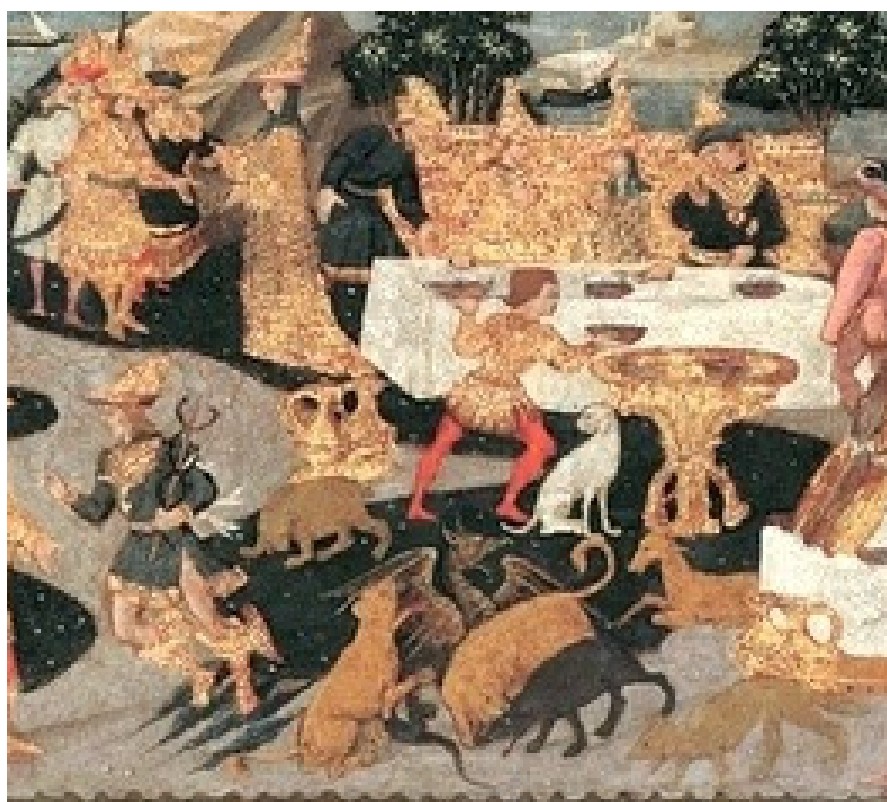

**Figure 5.** Detail, Figure 4.

It could be argued that the growling, bristling boars in this passage refer to Odysseus'
men, but characterizing them in this way removes them from the *Odyssey* narrative as
decisively as if they had never been mentioned. Not only are the metamorphosed Greeks
in Homer and Ovid depicted as meekly despairing, but their ultimate purpose is to set
the scene for Odysseus' heroic defeat of Circe, the accomplishment of which depends on
his success in convincing her to transform them back into their human shapes. Boccaccio
continues to follow Virgil in his *De mulieribus claris* account of Circe, where, never mention-
ing pigs or boars, he observes that the hero's companions, like sailors before them, "were
changed into various kinds of animals" (*Dmc* 38.3).

Apollonio's decision to not only eliminate all reference to swine but to replace Homer's
other harmless creatures with Virgil's (and Boccaccio's) intimidating beasts may have been
an acknowledgment that Renaissance readers had far greater first-hand familiarity with
the latter. As these vicious creatures are Apollonio's only signifiers of Circe's sorcery, it is
likely he considered them to be a distillation of the menace she presents. That Apollonio
would have turned to Virgil when rendering Homer's epic is supported by iconographical
decisions made in the depiction of Polyphemus in both the Kraków painting and in his
extremely condensed, one-panel version of Odysseus's adventures now in the Chicago
Institute of Art (Figure 6). For example, in a scene from the *Aeneid* in which the hero rescues
a man unwittingly deserted by the fleeing Greeks, Virgil describes "warm joints quivering
within [the cyclops's] jaws" (*Aen.* 3.812–13), intimating, as gruesomely represented in
Apollonio's images, that the Greeks were eaten alive.[21] Homer's cyclops, by contrast,
swiftly kills the men before dismembering them in preparation for his dinner.

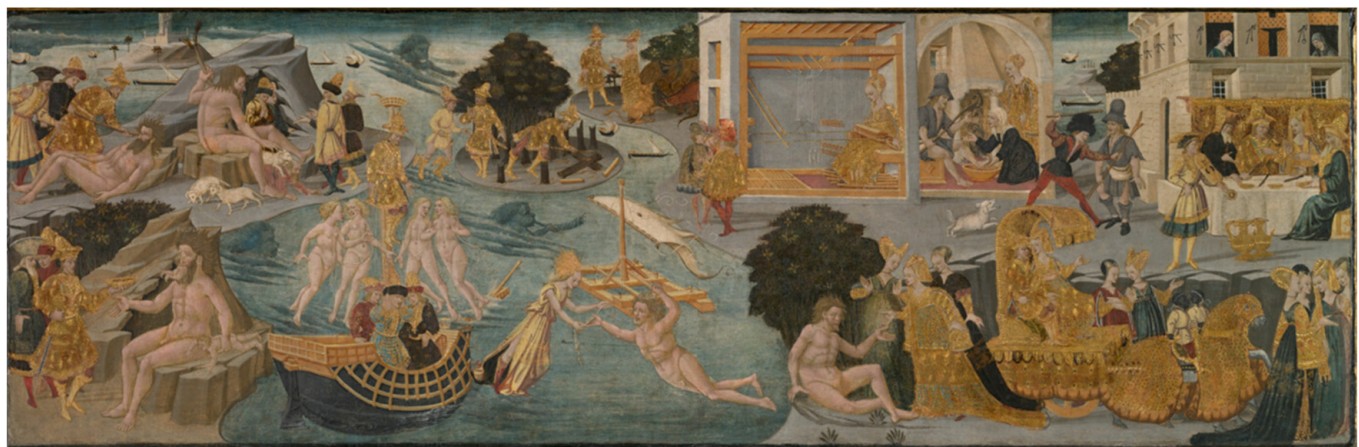

**Figure 6.** Apollonio di Giovanni, *The Adventures of Ulysses*, ca. 1435–1445, tempera on panel *cassone* front, Art Institute of Chicago, Mr. and Mrs. Martin A. Ryerson Collection, 1933.1006 (public domain photograph).

Odysseus and Circe assume essentially identical positions and postures when they meet in the Chicago and Kraków narratives (the central background of the former, the central middle ground of the latter). In the Chicago panel Odysseus is the only human figure in Circe's presence; her identity is again revealed by the wild beasts that surround her. In the Kraków panel Odysseus is recognizable by the hat, belt, boots, and gold garments he consistently wears, as well as by the prominent sword on which his hand rests.[22] Homer, Ovid, and Boccaccio all underscore the importance of Odysseus' sword to the successful subjugation of Circe's will. In the *Odyssey*, Hermes says "As soon as Circe with her long wand strikes you, then drawing back from beside your thigh your sharp sword, rush forward against Circe as if you were raging to kill her" (*Od*. X.293–295). Odysseus obeys: "She struck me with her wand … but I, drawing from beside my thigh the sharp sword, rushed forward against Circe" (*Od*. X.319, 321–322). Ovid writes that "when she tried to stroke his hair with her wand, he thrust her back, and frightened off the terrified goddess with drawn sword." (Ovid 319) As Apollonio's Circe has no wand, and the companions who cluster behind Odysseus are human, this part of his narrative aligns most closely with Boccaccio's: "When Ulysses drew his sword and threatened the sorceress with death, she changed his companions back to their original form" (*Dmc* 38.4). In the context of the *Odyssey* narrative, her outstretched arms can be read as a gesture signifying the success of Odysseus' intimidation tactics.

Like Boccaccio who, in keeping with his claim that Circe was a fantastical iteration of a common variety of disreputable woman, makes no mention of potions or wands, Apollonio's Circes are stripped of the accoutrements of witchcraft. The intimation of comparably potent phallic symbols is thus avoided; the painter's juxtaposition of Odysseus' sword with Circe's manifestly empty hands conveys her vulnerability in the face of his determined resistance. In the context of a story adapted to guide a Renaissance couple through married life, the painter's choice to depict a hero prepared to draw a sword on an unarmed patrician woman may best be understood as the show of virility necessary to withstand female encroachment on male prerogatives. Her general appearance aids in this reading; she looks nothing like Homer's "goddess with the glorious hair" (*Od*. X.220–221, 310–311), Virgil's "savage goddess" (*Aen*. VII.23), or Ovid's "dread goddess," (Ovid 318) but rather like an attentive hostess welcoming her husband and guests.

Apollonio further normalizes the goddess by stripping away the aura of opulent profusion with which Circe and her realm are imbued in the *Odyssey*. Homer describes a palace of "well-polished" stone (*Od*. X.211); "shining doors" (*Od*. X.230); and halls in which the Greeks, served by nymphs "born of the springs and from the coppices and the sacred rivers" (*Od*. X.350–351), feasted on "unlimited meat and sweet wine" for a year

(*Od*. X.468). There is no suggestion of supernatural abundance in the painter's rendering, where Circe's banquet for Odysseus is essentially identical to that of the Phaeacians' both on the far right of the Chicago panel and the central middle-ground of the Kraków pendant panel (Figure 7). The choice to understate Circe's environment and thereby render the surroundings, possessions, and attendants of a goddess indistinguishable from those of a wealthy woman reinforces a reading of her world as relevant to the lives of *cassone* viewers. Apollonio's omissions and modifications demystify Circe to the point where her interactions with Odysseus model those of a wife solicitous of her husband's comfort and approbation. The couple's complete lack of engagement with the wild beasts reinforces this portrait of genteel conventionality—and yet they continue to prowl and roar, an insidious reminder of Circe's potential to destabilize the social and political bedrock which she supports only under duress.

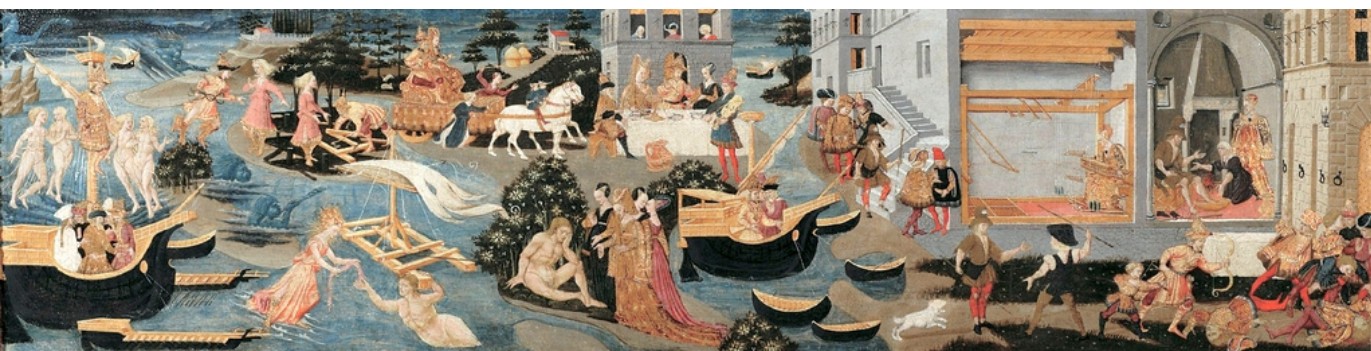

**Figure 7.** Apollonio di Giovanni, *The Adventures of Odysseus*, ca. 1430–1440, tempera on panel *cassone* front, Kraków, Royal Wawel Castle (public domain photograph).

Additional evidence of Circe's possible duplicity in assuming the role of a respectably domesticated woman lies in the absence of the loom on which she was weaving when the Greeks first met her. Homer introduces Circe as "singing in a sweet voice as she went up and down a great design on a loom . . . delicate and lovely and glorious" (*Od*. X.221–223). In *cassone* paintings she does not engage in this traditional woman's work (which she is also denied by Latin sources); the large looms that feature in both narratives are reserved for Penelope, Odysseus' archetypal good wife. In both the Chicago panel and the Kraków pendant, Penelope is encaged in loom-filled rooms made extremely salient by their unadorned simplicity. In the Chicago painting, the cloistered wife is positioned next to, and thereby visually juxtaposed with, Circe, who is surrounded by wild animals rather than thick walls.[23] In the Kraków pendants, the viewer is invited to compare-and-contrast the two women by their comparable positioning on the two panels. In both cases, Circe's close interactions with a man who is not her husband highlight Penelope's diligent marital constancy. Apollonio's elimination of Circe's female attendants/chaperones, prominent figures in both Homer and Ovid, is also suggestive of her continued disavowal of propriety.

In juxtaposing the enigmatic Circe with a categorically dependable wife, Apollonio magnifies both the obscure nature of the former's motives and the mutability of her behavior. The suspicion that she is masking deceit with a cultivated veneer is further amplified by the painter's decision to associate her story with that of Polyphemus. The Kraków panel suggests that these two adventures are temporally proximate, which is not the case; in Homer, Odysseus and his men contend with many perils after escaping from the cyclops but prior to meeting the sorceress. The omission of these other encounters allows the painter to construct a visual contrast between Polyphemus' indisputable barbarity (he is unclothed, shaggy-headed, lives in a cave, and eats people) and the semblance of civility signaled by Circe's decorum.[24] Odysseus' strategies for vanquishing first Polyphemus and then Circe are contrasted in propinquant episodes placed in the center foreground of the panel. Although the cyclops presents as the deadlier foe, Odysseus, an exceptional

man who understands male weakness, bests him through physical prowess and native wit (he inebriates then blinds him). Having already taken advantage of Trojan greed and gullibility with a wooden horse full of soldiers, the hero now uses his innate gifts to exploit Polyphemus's vulnerabilities. Left to his own devices, however, this shrewd master of trickery is defenseless against Circe's stratagems. Unlike Polyphemus, she doesn't rob her victims of life, but rather the birthright, as men, to act rationally in their own interests. The wisdom to withstand her must be acquired from a sympathetic divine source, whom Apollonio positions as a barrier between Odysseus and the fate he would otherwise suffer at her hands.

Renaissance artists repeatedly turned to both Virgil and Boccaccio for *cassone* narratives that revealed the danger female autonomy posed to the realization of male ambition. Many painters, including Apollonio, explored the relationship of Aeneas and Dido, the queen whose desire to make the hero her consort imperils his quest to found Rome. He escapes this potential snare and is shown in yet another Apollonio panel wedding Lavinia, a woman who unhappily submits to the marriage that will authenticate his right to rule. The wedding ceremony in this painting is juxtaposed with the gory death of Aeneas's virgin-warrior antagonist, Camilla.[25] Boccaccio's *Teseida delle nozze d'Emilia* inspired Apollonio and others to paint the spectacle of other trampled, bloody Amazons battling against male sovereignty. The price of their humiliation at the hands of clearly superior male warriors includes acquiescing to the victors' marriage proposals (see Franklin 2010). The necessity of a woman's acceptance of subjugation to a husband's will was sometimes communicated with an even heavier hand. Sandro Botticelli was commissioned to paint a series of *spalliera* panels depicting the tale of Nastagio degli Onesti, made popular in Boccaccio's *Decameron* (V.8), in which a proud young woman agrees to marry a persistent suitor only after witnessing the divinely-ordained punishment of a similarly defiant woman: after their deaths, the rejected wooer repeatedly hunts down the object of his love, who is then mauled by his dogs.[26] Botticelli also painted *spalliera* narratives of the exemplary Griselda (*Decameron* X.10), who for years suffered without complaint the cruel torments meted out by her husband and was ultimately rewarded with his approbation.[27]

Throughout *De mulieribus claris* Boccaccio calls for the diligent defense of the prerogatives of men against the machinations of women. Circe is but one of his famous women who embody the female compulsion to emasculate men, causing them to sacrifice their authority, rationality, and reputations to malignant female ambition. Iole was another; her husband Hercules "fell into an appalling state of subjection" (*Dmc* 23.15, p. 97) because Iole believed there was "glory in weakening a strong man" (*Dmc* 23.4, p. 93). Boccaccio closes this account with the following harangue: "a strong and powerful enemy threatens us, and those concerned for their own well-being should be very much afraid and rouse themselves out of their indifference." This fear is reinforced in his *De casibus virorum illustrium*, where he writes, "Women have complete contempt for the laws of God ... they try to achieve sovereignty by a sort of inborn diligence." (Boccaccio 1965, p. 41). The innately unreliable and even pernicious nature of women and the consequent need to closely circumscribe and monitor their activities also permeate quattrocento humanist-penned conduct manuals.[28] A warning to new husbands (and future sons) not to cede power to women may therefore account for the compositional prominence of Apollonio's wild beasts—neither specific to nor embedded within Odysseus's narrative, the creatures can be read as representing a more generalized threat posed by women who seek to control men.[29]

Although Homer's account delineates the many ways in which Circe's knowledge and resources enable the foundered hero to continue on his homeward journey, there is no suggestion of his reliance on her multifarious assets in *cassone* paintings.[30] Yarnell observes that, in Virgil, "there is no hint that Aeneas has anything to learn" from Circe, who "represents the archetypal feminine projected as evil and then evaded, safely skirted by." (Yarnall 1994, p. 82). Apollonio's Circe, visually indistinguishable from a patrician wife, has nonetheless succeeded in subverting the natural order by transforming men into beings that look fierce but, captive to a woman, are functionally inert. Her apparently gracious

assumption of household duties may be achieved through artifice and overlay both the will and the capacity to wreak havoc on men's lives. The wise man will anticipate and defend against her schemes, making it clear who's boss and enjoying a good dinner before heading off on his next adventure.

**Funding:** This research received no external funding.

**Data Availability Statement:** Not applicable.

**Conflicts of Interest:** The author declares no conflict of interest.

## Notes

[1]   See (Sowerby 1997; Wilson-Okamura 2010, esp. pp. 124–32). For Homer's perceived value as a source of enlightenment in the Renaissance see (Poliziano 2007; Ford 2007).

[2]   For Pilato and the humanists see (Kircher 2014; Botley 2004;). Sowerby (1997, p. 186) notes that Pier Candido Decembrio's partial revision (c.1440) and Lorenzo Valla's prose translation of the *Iliad* (c.1444) did little to remedy Pilato's inadequacies.

[3]   All *Aeneid* translations are from (Mandelbaum 1971).

[4]   Scholarship relevant to the treatment of Circe by Virgil and Ovid is vast; for bibliography and analysis see, for example, (Yarnall 1994, esp. pp. 79–91). For an expansive essay on Circe and transformation that addresses the work of other ancient writers (Plutarch, Pliny), see (Warner 1997).

[5]   For early quattrocento use of Diodorus see (Robathan 1932).

[6]   For the popularity of *De mulieribus claris* and Boccaccio's other scholarly work during the Renaissance see, for example, (Bec 1984, esp. pp. 27, 112–14; Bozzolo 1973, pp. 22–43; Tournoy 1977).

[7]   All *De mulieribus claris* translations are by (Brown 2001).

[8]   For ancient hybrid imagery in the Circe story see (Brilliant 1995; Buitron and Cohen 1992). Di Febo (2019, p. 111) interprets the hybrid imagery in this miniature as visualizing Circe's power to ensnare men by robbing them of their reason. For this miniature see also (Desmond and Sheingorn 2001, pp. 18–19).

[9]   See (Callmann 1974, p. 17; Miziolek 2006, p. 66). Apollonio's paintings are dated decades prior to the woodcut of Circe used in many editions of *De mulieribus claris* printed in or after 1473. Unlike earlier French manuscript paintings, the Circe in these prints is modestly clothed and, without the identifying inscription, is indistinguishable from a human woman. It is perhaps noteworthy that Odysseus himself was never celebrated as an exemplar in contemporary *uomini famosi* (famous men) compendia or portrait cycles.

[10]   For late medieval and early Renaissance views on occult magic see, for example, (Bailey 2017; Kieckhefer 2013). The conception of witchcraft as demonic power exercised by humans who had relinquished their souls to the devil only began to develop into a coherent theory in the middle decades of the 15th century; see, for example, (Duni 2007).

[11]   The multi-disciplinary scholarship relevant to witchcraft in the Early Modern period is vast; foundational is, for example, (Russell 1972) and (Cohn [1975] 2000); see also (Stephens 2001). Interest in, and persecution of, witches was fueled by the publication of works such as the *Malleus maleficarum* (*Hammer of Witches*) in 1486; see (Broedel 2003). For Circe in the sixteenth-century visual arts see (Zika 2007, pp. 133–55; Yarnall 1994, pp. 99–144).

[12]   Seminal for *cassone* studies is (Schubring 1923); and, for Apollonio, (Callmann 1974).

[13]   Giorgio Vasari, sixteenth-century painter, biographer, and historian, describes the decoration of *cassoni* as including "fables taken from Ovid and other poets, or rather stories told by the Greek and Latin historians" (Vasari 1878, pp. 148–49)

[14]   For the later, less accomplished work, see (Callmann 1974, p. 17).

[15]   On the far right of the panel Odysseus' ship leaves Circe's island and the men have an encounter with the monstrous Scylla.

[16]   For the long association of Hermes and the caduceus with one or more of these qualities see, for example, (Freedman 2011; Schlam 2009, p. 15; Friedlander 1992; Shelmerdine 1986, pp. 49–63). For moly see, for example, (Clay 1972; Stannard 1962).

[17]   All *Odyssey* translations are by (Lattimore 1967).

[18]   There is also a stag and a white dog; it is not clear whether they are intended to be read in conjunction with the beasts or with the servant.

[19]   All *Metamorphoses* translations are by Mary M. Innes (Ovid 1955).

[20]   This is Mandelbaum's translation of Virgil; Jon Solomon (Boccaccio 2011) translates this passage in the *Genealogia deorum gentilium* as "From here the angry roaring of lions battling their bonds and bellowing under the late night, and bristly swine and bears in their pens rage, and the bodies of great wolves howl—once human in appearance, the cruel goddess with powerful herbs, Circe, cloaked them with the faces and backs of beasts."

[21]   For more evidence of Apollonio's use of Virgil's characterization of the Polyphemus story see (Franklin 2018; Miziolek 2006). For the inclusion of non-Homeric material in Apollonio's *Odyssey* panels that was introduced to resonate specifically with Early Renaissance viewers see (Zerba 2017).

[22]   He should not, therefore, be mistaken for the leader of the doomed reconnaissance party, none of whom attempts to intimidate Circe.

[23]   It is worth noting that the loom was not used for purposes of contradistinction in antique vase paintings of Circe and Penelope, where both women were frequently depicted with this archetypal domestic object. See, for example, (Brilliant 1995, pp. 171–72).

[24]   Apollonio renders the appearances of both Polyphemus and Circe more human than does Homer, a change that presumably made Odysseus' fantastic adventures, and the lessons to be derived from them, more relatable to quattrocento viewers. Homer, for example, says that the cyclops is a "monstrous wonder . . . not like a man . . . but more like a wooded peak of the high mountains" (*Od.* IX.190–92).

[25]   For this and other Camilla/Lavinia *cassone* narratives see (Franklin 2013; Baskins 1998, pp. 75–102).

[26]   *Spalliere* were larger than *cassoni* and made to set into a wall above a bed or other piece of furniture. See (Olsen 1992).

[27]   For Renaissance Griselda imagery see (Baskins 1991).

[28]   See, for example, Leon Battisti Alberti, *I libri della famiglia*, c.1430; and Francesco Barbaro, *De re uxoria liber*, 1416.

[29]   Scholarship addressing the nature, roles, and expectations of (most) upper-class women, especially those living in quattrocento republics (as opposed to those groomed to be the consorts of princes) is vast. See, for example, (Panizza 2000; Wiesner 1993; Klapisch-Zuber 1988). Of course, there were exceptional women who achieved measures of public success in spite of legal, political and social strictures; see, for example, (Robin 2013).

[30]   In addition to building and stocking a ship for Odysseus and his men, Circe instructs him on withstanding the Sirens, warns him of the dangers presented by Scylla and Charybdis; admonishes him not to harm the cattle of the sun on Thrinacia; and directs him to seek the advice of Teiresias in the underworld.

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
