# Peer review of "Transforming Circe: Latin Influences on the Depiction of a Sorceress in Renaissance Cassone Narratives"

_arts, 2023_

Round 1

Reviewer 1 Report

Partly because the illustrations show the entire cassone panel, the visual evidence tends to seem secondary to the literary sources. If it were possible to include a few details, the point about Apollonius' not using pigs as the beasts Circe makes could be make more visually compelling and possibly worth more examination, rather than only reinforcing the importance of Virgil's account. In general, could the illustrations, i.e., the works of art, bear more of the argument's weight?  Also, the second part of footnote 39 seems a last minute addition (lacks italics, spacing is irregular), and yet the positive views of women certainly deserve attention, too?  As written, the article confirms the biases most readers will bring to the subject. And yet, I am convinced that Boccaccio and others had more complex ideas about gender than anyone would learn from this article. E.g., to take Griselda as a straightforward exposition of wifely submission is scarcely appropriate. Surely she is a female Job, an allegorical figure of patience and faith well outside of any project of realistic representation. The article would be so much more valuable if its basic thesis were not taken as a foregone conclusion, but treated with nuance. And the issue could be addressed to the paintings.  Is there any ambiguity in presenting female protagonists, or are they merely cardboard seductresses or failed warriors, when they are not dutiful wives or the Virgin? Do the texts outdo the images in making the female villains and losers complicated? The Nastagio story, after all, has to do with social class as well as gender politics. How does the presentation of female figures change as pictures in general become more sensuous (there is a late sixteenth-century woodcut included, and by then the representation of flesh was a different ballgame than in the fourteenth century)? And does that development itself change how artists illustrate texts? It certainly changes how they represent the Virgin.

Author Response

I agree that details of the cassone images would be useful and hope it will be possible to incorporate more into the text of this article.  The formatting of footnote 39 will definitely be corrected.  I agree that scholars should dig evermore deeply into the significance of literary and visual treatments of a wide variety of women, and many interesting avenues of exploration are suggested by this reviewer.  

Reviewer 2 Report

I have nothing to criticize here, nor anything to suggest. This is a clearly-written, insightful, and well-supported reading that makes sense of what on the face of it is an odd invocation of Circe. Excellent work, imo.

Author Response

Thank you so much for your time and the supportive review.

Reviewer 3 Report

I love this article! It is a well-written study on the depictions of Circe on Renaissance cassoni (marriage chests). The author argues convincingly that while Circe is shown as a powerful being able to transform men into beasts, her depictions in the Renaissance wedding context served as warnings to grooms against women who seek to control me.  

The arguments about transformation/hybridity reminded me of Caroline Bynum's work on Metamorphosis and Identity. It seemed to me that this article on Circe jumps from Classical literature directly to a Renaissance understanding of Classical literature, but I wondered if the Renaissance world had formed any views inherited from the middle ages. For example, on p. 4 the author states that the only known late trecento/early quattrocento visual depictions of Circe are manuscript miniatures. A note about what other visual depictions of Circe from the early Middle Ages may have (or may not have) preceded this material would be nice. Was this theme not of interest in the Middle Ages? Were the visual narratives on the Renaissance chests only indebted to the literary Classical accounts?

This is just a passing thought.  The article stands on its own!

I noticed a few minor things in the footnotes. For example, the spacing is strange on Note 39 and the Journal should be italicized. I just recommend a quick edit.

Author Response

Thank you so much for your time and your supportive review.  I was not able to find any earlier European medieval Circe imagery; I agree that, if it exists (or ever existed), it would be important to consider!